# Glucose Catabolite Repression Participates in the Regulation of Sialidase Biosynthesis by Antarctic Strain *Penicillium griseofulvum* P29

**DOI:** 10.3390/jof10040241

**Published:** 2024-03-23

**Authors:** Radoslav Abrashev, Ekaterina Krumova, Penka Petrova, Rumyana Eneva, Vladislava Dishliyska, Yana Gocheva, Stefan Engibarov, Jeny Miteva-Staleva, Boryana Spasova, Vera Kolyovska, Maria Angelova

**Affiliations:** 1Department of Mycology, The Stephan Angeloff Institute of Microbiology, Bulgarian Academy of Sciences, Academician G. Bonchev 26, 1113 Sofia, Bulgaria; rabrashev@abv.bg (R.A.); ekrumova@abv.bg (E.K.); vladydacheva@yahoo.com (V.D.); j_m@abv.bg (J.M.-S.); bkasovska@abv.bg (B.S.); 2Department of General Microbiology, The Stephan Angeloff Institute of Microbiology, Bulgarian Academy of Sciences, Academician G. Bonchev 26, 1113 Sofia, Bulgaria; pepipetrova@yahoo.com (P.P.); rum_eneva@abv.bg (R.E.); yana2712@gmail.com (Y.G.); stefan_engibarov@abv.bg (S.E.); 3Institute of Experimental Morphology, Pathology and Anthropology with Museum, Bulgarian Academy of Sciences, Academician G. Bonchev 25, 1113 Sofia, Bulgaria; verakol@abv.bg

**Keywords:** Antarctic fungi, sialidase, cold-active enzyme, catabolite repression, cAMP, glucose-6-phosphate, sialidase gene expression

## Abstract

Sialidases (neuraminidases) catalyze the removal of terminal sialic acid residues from glycoproteins. Novel enzymes from non-clinical isolates are of increasing interest regarding their application in the food and pharmaceutical industry. The present study aimed to evaluate the participation of carbon catabolite repression (CCR) in the regulation of cold-active sialidase biosynthesis by the psychrotolerant fungal strain *Penicillium griseofulvum* P29, isolated from Antarctica. The presence of glucose inhibited sialidase activity in growing and non-growing fungal mycelia in a dose- and time-dependent manner. The same response was demonstrated with maltose and sucrose. The replacement of glucose with glucose-6-phosphate also exerted CCR. The addition of cAMP resulted in the partial de-repression of sialidase synthesis. The CCR in the psychrotolerant strain *P. griseofulvum* P29 did not depend on temperature. Sialidase might be subject to glucose repression by both at 10 and 25 °C. The fluorescent assay using 4MU-Neu5Ac for enzyme activity determination under increasing glucose concentrations evidenced that CCR may have a regulatory role in sialidase production. The real-time RT-PCR experiments revealed that the sialidase gene was subject to glucose repression. To our knowledge, this is the first report that has studied the effect of CCR on cold-active sialidase, produced by an Antarctic strain.

## 1. Introduction

Sialidases (EC 3.2.1.18, acetylneuraminyl hydrolases, exo-alpha sialidases, and neuraminidases) are a large family including enzymes that catalyze the removal of terminal sialic acid residues from glycoproteins. They are involved in different physiological processes such as cell proliferation, immune function, and pathophysiological conditions (human cancers and infections) [1]. Microbial sialidases have an important role in nutrition, adhesion, and invasion host cells. The most recognized producers of sialidase are pathogenic bacteria such as *Clostridium perfringens*, *Streptococcus pneumonia*, *Vibrio cholerae*, and *Streptococcus* spp. (group A and B), the causative agents of gangrene, pneumonia cholera, meningitis, and glomerulonephritis, respectively [1]. This enzyme has been detected in *Pasteurella multocida* (zooanthroponosis), *Corynebacterium diphtheriae* (diphtheria), *Salmonella typhimurium* (salmonellosis), etc. [2,3]. Influenza virus neuraminidase is the best-known viral enzyme. It is involved in the recognition and modulation of sialic acids on the cell surface as the virus receptor [4]. However, the reports on sialidase presence in fungi are very rare. There is information on sialidase synthesis in several *Aspergillus* species that causes invasive aspergillosis [5,6,7] and pathogenic strains of *Candida albicans* isolated from a human vaginal specimen [8].

Purified sialidase preparations can be used for virus receptor studies, cell hybridization, and interaction of lymphocytes with tumor cells, analysis of oligosaccharides, glycoproteins, and glycolipids, or for the design of new drugs and therapies [9,10,11]. One promising direction of sialidase application is the use of the enzyme to inhibit viral infection. According to Giurgea et al. (2020) [12], supplementing current vaccines against the influenza virus infection with sialidase preparations could be an encouraging strategy. Sialidase isolated from the bacterium *Clostridium perfringens* (Sialivac) can prevent the spread of the influenza virus infection in livestock [13]. Similar results have been demonstrated for sialidase from *Actinomyces viscosus* [3]. Finding a new sialidase is of increased interest, particularly one from nonvirulent fungi for use in the pharmaceutical and food sectors since secreted enzymes can be easily overexpressed and purified in large quantities from a fungal culture [14,15].

As is known, enzyme production is under control by mechanisms of induction and repression. The expression of genes is induced by the presence of particular substrates, while the addition of glucose to the medium represses the synthesis of certain enzymes. Therefore, understanding the regulation of sialidase can be both scientifically and industrially important. The study of the control mechanisms regulating the synthesis of microbial sialidases dates back a long time. One of the first studies was that of Wang et al. (1978) [16] on the induction and repression of *Arthrobacter sialophilus* sialidase. Carbon catabolite repression (CCR) acts when the cells grow in a medium containing two or more carbon sources, preferring to utilize the easily metabolizable one (frequently glucose) while at the same time repressing the genes responsible for the use of alternative carbon sources. This mechanism participates in the regulation of sialidase synthesis by different microbial cells. Quantitative gene expression analysis of *S. pneumoniae* demonstrated repression of the locus coding for the two main pneumococcal sialidases by high concentrations of glucose [17]. Similar data have been reported for the *C. perfringens* producer of three sialidases, coding by *nanJ*, *nanI*, and *nanH* genes [1,18,19]. The expression of *nanI* decreases in the presence of high glucose concentrations. According to Blanchette et al. (2016) [20], pneumococcal neuraminidase A (NanA) is a factor for biofilm formation that was inhibited by glucose, sucrose, and fructose.

CCR is also used by fungi for precise adaptation of their metabolism to the environment. There are many reports on glucose repression of fungal amylase [21], xylanase [22], hemicellulase [23], lignocellulolytic enzymes [24], pectin lyase [25], etc. However, studies of the glucose effect on fungal sialidase synthesis have not been found.

Our previous study reported the distribution of sialidases among 113 fungal strains from non-clinical isolates taken from different habitats, including Antarctica [26]. The sialidase gene was identified in sialidase-positive and sialidase-negative fungal strains. The Antarctic strain *Penicillium griseofulvum* P29 was selected as the most promising producer. The purified *P. griseofulvum* P29 sialidase was characterized as a cold-active enzyme [27]. Cold-active enzymes exhibit maximum activity at low ambient temperatures, characterizing them as attractive biocatalysts for medical and industrial applications [28]. However, despite the increasing attention on sialidases, no information regarding the regulatory mechanisms of cold-active sialidase has been published.

In this context, the present study aimed to evaluate the participation of CCR in the regulation of cold-active sialidase biosynthesis by the Antarctic strain *P. griseofulvum* P29. To our knowledge, this is the first report on the glucose effect on temperature-sensitive sialidase.

## 2. Materials and Methods

### 2.1. Fungal Strains and Culture Conditions

All fungal strains used in this study, *P. griseofulvum* P29, *P. fimorum* III 7-1, *Aspergillus tubingensis* A 3-1, and *A. niger* A 3-2, belong to the Mycological collection of the Stephan Angeloff Institute of Microbiology, Bulgarian Academy of Sciences. They were isolated as follows: *P. griseofulvum* P29 and *P. fimorum* III 7-1 from soil samples taken from Terra Nova Bay and Livingston Island, Antarctica, respectively; *A. tubingensis* A 3-1 and *A. niger* A 3-2 from soil samples taken from Denali National Park, Alaska [26]. *P. griseofulvum* P29 was deposited at the National Bank for Industrial Microorganisms and Cell Cultures, Bulgaria (NBIMCC 9106).

Long-term preservation was carried out in the Microbank system (Prolab Diagnostics, Richmond Hill, Canada) consisting of sterile vials that contained 25 porous, colored beads and a cryopreservative fluid at −80 °C. Before use, the conidiospores were grown on Beer agar at 28 °C for 7 days. *P. griseofulvum* P29 was characterized as a psychrotolerant strain (with optimal growth temperature of 25 °C) producer of a cold-active sialidase [26,27].

The composition of the culture medium Wh used for submerged cultivation was as follows (g per liter): whey, 20.0; ammonium citrate, 7.5; KH_2_PO_4_, 1.0; MgSO_4_·7H_2_O, 0.5; KCl, 0.5; FeSO_4_·7H_2_O, 0.5; MnSO_4_, 0.0025; CuSO_4_, 0.0011. For the experiment of CCR, the medium was supplemented with glucose (0.05, 0.1, 0.5, 1.0, 2.0, 3.0, and 5.0%), sucrose (2.0%), maltose (2%), or glucose-6-phosphate (2%). Whenever required, cAMP (2, 5, and 8 mM) was added to the medium containing 2% glucose.

Both growing and non-growing mycelial systems were used. For the growing mycelia, 74 mL of corresponding medium were inoculated with 6 mL spore suspension at a concentration of 2 × 10^8^ spores/mL in 500 mL Erlenmeyer flasks. The cultivation was performed on a rotary shaker (220 rpm) at 25 °C for 24, 48, and 72 h.

To prepare non-growing mycelia, the 24 h cultures on the Wh medium were filtered through a Whatman (Clifton, NJ, USA) No 4 filter, and the sialidase activity of the filtrate was measured. Separated mycelia were washed with potassium phosphate buffer, excess water was removed by pressing between dry filter papers, and they were divided into portions of 2 g. Each portion was added to 40 mL of corresponding medium in a 300 mL Erlenmeyer flask, followed by incubation for 2, 4, and 6 h at 25 °C on a rotary shaker (220 rpm). The incubation was carried out in the presence of chlortetracycline (10 μg/mL) to prevent bacterial contamination.

### 2.2. Enzyme Activity Determination

The sialidase activity was measured quantitatively by colorimetric determination of free sialic acid by the thiobarbituric acid method of Uchida et al. (1977) [29]. One unit of enzyme activity was defined as the amount that releases 1 μg of N-acetyl-neuraminic acid for 1 min at 37 °C, using glucomacropeptide [30] as a substrate.

Culture filtrates from *P. griseofulvum* P29 grown in the absence or presence of 2 or 5% glucose were screened qualitatively for sialidase in broth-spot fluorescent assays, as was described by Lichtensteiger and Vimr (1997) [31]. As a fluorescent substrate, 2′-(4-methylumbelliferyl) α-D-N-acetylneuraminic acid (4MU-Neu5Ac; Sigma-Aldrich, Buchs, Switzerland) was used. Sialidase-positive supernatants were detected by their intense green-blue fluorescent halos when excited with 366 nm light (UV-Box DESAGA Sarstedt-Gruppe, Wiesloch, Germany). As negative controls, substrate 4MU-Neu5Ac alone and supernatants without 4MU-Neu5Ac were used.

### 2.3. RNA Isolation, Copy-DNA Synthesis, and Real-Time RT PCR

Total RNA was isolated from 0.2 g biomass of the fungal strain cultured in media containing either glucose (as a potential repressor of the transcription) or 2% whey. The biomass was grounded in a mortar with a pestle after freezing with liquid nitrogen, the mixture was re-suspended in 500 μL TE buffer (10 mM TRIS/HCl pH 8.0, 1 mM EDTA Sodium salt) and centrifuged at 10,000× *g* for 5 min, at 4 °C. The resulting supernatant (~300 μL) was used for RNA isolation.

Reverse transcription was performed with the NG dART RT Mix (EURx, Gdansk, Poland) in 20 μL reactions with 1 μg total RNA, 200 ng random hexamer, and the following temperature regime: 10 min at 25 °C for primer hybridization, 50 min at 50 °C for the reverse transcription itself, and 5 min at 85 °C for the inactivation of the enzyme. Before the reverse transcription, the RNA samples were treated with 5U DNase I in a buffer with 25 mM MgCl_2_ (EURx, Gdansk, Poland) for 30 min at 37 °C. The enzyme was inactivated for 10 min at 65 °C in the presence of 20 mM EDTA.

Real-time PCR (RT-PCR) was performed using SsoFast™ EvaGreen^®^ Supermix with Low ROX (BioRad, Hercules, CA, USA) in a Corbett Research RG-6000 RT-PCR Thermocycler (Qiagen, Germantown, MD, USA). Primer design was performed using the Primer-BLAST program [32] (Table 1). Primer pair Pen_RTF/Pen_RTR was based on the sequence of the sialidase gene of *P. griseofulvum* P29 (NCBI GenBank accession no. MT647999) reported previously [26]. It generated fragments of 100 bp. The glyceraldehyde-3-phosphate dehydrogenase gene was used as a housekeeping gene and was detected by primer pair GADPH_F/GADPH_R [33]. The real-time PCR mix contained 40 ng of cDNA as a template, and 500 nM of the primers, final volume 20 µL, and was carried out at an annealing temperature of 65° C.

Relative expression was calculated using the ∆∆Ct method as ∆Ct = Ct_gene_ − Ct_GADPH_. ∆∆Ct = ∆Ct_sample_ − ∆Ct_control_; fold Change (FC) = 2^−∆∆Ct^. The presented values were obtained as a mean from two independent cultivation experiments.

### 2.4. Analytical Methods

Soluble reducing sugars were determined by the Somogy–Nelson method [34]. The dry weight determination was performed on samples of mycelia harvested throughout the culture period. The culture fluid was filtered through a Whatman (Clifton, USA) No 4 filter. The separated mycelia were washed twice with distilled water and dried to a constant weight at 105 °C.

### 2.5. Statistical Evaluation of the Results

The results obtained in this investigation were evaluated from at least three repeated experiments using three parallel runs, and reported values represent the mean. The error bars indicate the standard deviation (SD) of the mean of triplicate experiments. The data were analyzed using one-way analysis of variance (ANOVA), followed by Tukey’s test. For the statistical processing of the data, the version of the ANOVA software built into Origin program (OriginPro 2019b, 64-bit) was used.

## 3. Results

### 3.1. Glucose-Mediated Repression of Sialidase Production by Growing Fungal Mycelia

Growing mycelia (24 h cultures) from four strains, *P. griseofulvum* P29, *P. fimorum* III 7-1, *A. tubingensis* A 3-1, and *A. niger* A 3-2, were monitored for the production of biomass and sialidase in Wh medium supplemented, or not, with glucose (Figure 1).

The growth demonstrated a significant increase in the presence of 2 and 5% glucose with comparison to the medium without glucose (Figure 1A). The addition of glucose led to about a 1.2- to 1.4-fold and 1.9- to 2.4-fold higher biomass content in the medium supplemented with a 2 or 5% glucose, respectively. At the same conditions, sialidase synthesis showed strong repression in the Wh medium containing glucose (Figure 1B). The presence of 2% glucose led to a 2- or 5-fold decrease in sialidase activity compared to the control cultures in the tested strains. However, the addition of 5% glucose resulted in a drastic decline in enzyme activity (about 12- to 18-fold) in all experiments.

Based on the results in Figure 1, it was evident that the strain *P. griseofulvum* P29 is a better producer of sialidase in the Wh medium. This fungal strain was selected for the next experiments. Table 2 shows the biomass content and sialidase activity in *P. griseofulvum* P29 grown in the Wh medium supplemented with 2% of different carbon sources. Significant growth on all monosaccharides and disaccharides after 24, 48, and 72 h of cultivation was observed. Cultures on the control medium (Wh) showed the highest enzyme activity level. Maximum activity (10 U/mL) was obtained after 24 h of cultivation. The addition of glucose, maltose, and sucrose caused significant repression of sialidase synthesis within the first 24 h. However, a decrease in sensitivity to repression occurred after 48 and 72 h of cultivation.

The dynamics of growth, enzyme activity, and residual sugar concentrations in the presence or absence of glucose in the medium are shown in Figure 2. The experiments included a wide range of glucose concentrations. An increase in biomass content was observed concomitantly with an increase in initial glucose concentration and time of cultivation (Figure 2A). The enzyme activity demonstrated an opposite trend (Figure 2B). Within the first 24 h of *P. griseofulvum* P29 cultivation, the addition of low glucose concentrations (0.05, 0.1, 0.5%) had a positive effect on sialidase production. However, the higher glucose concentrations (1 and 2%) resulted in a 1.3- and 2.1-fold decrease in enzyme activity, respectively. The subsequent enhancement of glucose content (3 and 5%) acts as a powerful inhibitor of enzyme synthesis; only traces of activity are measured under these conditions. It should be noted that in the next 48 and 72 h, attenuation of the effect of glucose concentrations was established. While in the presence of glucose up to 2% the activity almost equaled that of the control, with higher concentrations again strongly inhibited the synthesis.

In accordance with fungal growth, *P. griseofulvum* P29 displayed an active glucose uptake (Figure 2C). The fungal strain cultivated on the control medium or glucose at low concentrations (0.1% and 0.5%) completely assimilated the carbon source after 24 h of cultivation, while the addition of the highest glucose concentrations (3 and 5%) allowed high levels of residual sugars to be maintained even after 48 and 72 h. The ability of glucose to repress sialidase production by *P. griseofulvum* P29 was confirmed by the fluorescent assay against 4MU-Neu5Ac (Figure 3).

The culture filtrates from *P. griseofulvum* P29 mycelia grown on Wh medium with or without glucose demonstrated the presence or absence of green-blue fluorescent halos after UV light (λ 366 nm) irradiation. The results showed a difference in the intensity between the fluorescence of the broth spots depending on glucose addition. Intense fluorescence was established in the culture filtrate from Wh medium without glucose (sample 1). The addition of 2% glucose (sample 2) revealed very weak fluorescence that corresponded to the two-fold decrease in sialidase activity (see Figure 2B). As expected, sample 3 (Wh + 5% glucose) did not demonstrate fluorescent halos because this concentration strongly inhibited the activity.

### 3.2. Effect of Catabolite Repression on Non-Growing Mycelia

To prove the effect of catabolite repression on sialidase activity, the experiment was carried out with non-growing mycelia of *P. griseofulvum* P29. Figure 4 gives the results of the action of glucose on 24 h non-growing mycelia at 1, 2, 3, 4, and 5%. Non-growing mycelia in a medium without glucose were used as a control.

The enzyme activity decreased with increasing glucose concentrations during the whole period of incubation. After the first 2 h, the treatment with 1, 2, and 3% glucose resulted in 1.4-, 4.4-, and 6.0-fold lower activity in comparison to the control, respectively. The presence of 5% glucose almost completely inhibited sialidase synthesis (0.42 vs. 6.6 U/mL). After 4 and 6 h of incubation, the enzyme activity in the control increased from 6.6 to 7.6 and 9.3 U/mL, respectively. Variants with glucose addition maintained the diminished tendency of sialidase synthesis, especially for the high concentrations 2, 3, and 5%.

### 3.3. Influence of Catabolite Repression Regulators

To determine which other regulators are involved in the glucose-mediated repression of sialidase activity in the non-growing fungal mycelia of *P. griseofulvum* P29, the effect of glucose-6-phosphate and cAMP was investigated. As can be seen in Figure 5, exogenously added glucose-6-phosphate instead of glucose exerted repression in sialidase activity comparable to that caused by glucose. The reduced enzyme activity was maintained throughout the 6 h incubation period.

*P. griseofulvum* P29 mycelium was transferred from a Wh medium to a medium with 2% glucose in the absence or presence of cAMP. The exogenous cAMP also partially de-repressed the glucose effect on the sialidase activity (Figure 6). The obtained results showed a dependence on the concentration of cAMP and the time of incubation. Non-growing mycelia demonstrated an increase in activity from the 2nd to the 6th hour. Furthermore, the enhanced amount of cAMP (2, 5, 8 mM) correlated with two-fold higher enzyme activity compared to the glucose variant.

### 3.4. Comparative Sialidase Gene Expression in the Presence of Glucose

To prove the role of glucose in the regulation of sialidase synthesis, we quantified gene expression during growth after 24 and 48 h of cultivation in the presence of glucose (1, 2, and 5%) (Figure 7). The results were compared with those of the mycelium grown without glucose.

A significant decrease in transcription in the presence of glucose in a dose- and time-dependent manner was determined. After 24 h of cultivation, sialidase gene expression was down-regulated by the presence of glucose. The addition of 1, 2, and 5% glucose in the medium resulted in a 1.5, 1.7, and 2.22-fold decrease in sialidase gene expression, respectively. Interestingly, a significantly weaker effect on the sialidase gene transcriptional level after 48 h was evaluated, probably due to the decrease in the glucose level in the medium. Mycelia grown on medium with initial glucose concentrations of 1 and 2% showed gene expression levels approaching that of the control. However, at 5% glucose, even after 48 h of cultivation, down-regulation of the sialidase gene expression by ~30% was still observed.

### 3.5. Effect of the Temperature on Catabolite Repression of Cold-Active Sialidase

The sialidase produced by *P. griseofulvum* P29 was defined as a cold-active enzyme that could be produced at a wide range of temperatures, from 10 to 30 °C [26,27]. It is of interest to investigate the role of catabolite repression in enzyme synthesis at lower cultivation temperatures. The results of the experiments at 10 and 25 °C described the same response to glucose presence (Figure 8). A glucose concentration of 2% caused about 78 and 73% decrease in sialidase activity at 10 and 25 °C, respectively. The addition of 5% glucose almost completely repressed enzyme synthesis (about 95%) at both cultivation temperatures. The replacement of glucose by glucose-6-phosphate caused also about an 80% decrease in sialidase activity regardless of growth temperatures.

## 4. Discussion

Bacterial sialidases are among the most studied glycoside hydrolases due to their role in pathogenicity [1,3]. There is also information about the participation of the control mechanisms regulating their syntheses, such as induction and repression [1,17,19,35]. In contrast, studies of fungal sialidases have only begun in the last decade, and the mechanisms of regulation of gene expression and enzyme synthesis and activity remain completely unclear. However, the understanding of these mechanisms is crucial for scientific research as well as the development of effective methods for sialidase production.

As in bacteria, catabolite repression in fungi is a common mechanism of gene expression regulation that prevents the synthesis of enzymes and secondary metabolites unnecessary for the cell under certain conditions. CCR is exerted on many genes encoding glycoside hydrolases such as fungal β-1,3-glucanase and β-1,6-glucanase [36], intracellular β-glucosidase in *P. verruculosum* [37], or the penicillin-producing enzyme cluster in *P. chrysogenum* [38].

The main finding of the present study is that glucose catabolite repression plays a significant role in sialidase synthesis by *P. griseofulvum* P29. First, sialidase activity can be dramatically repressed by adding glucose, although its presence in the medium caused a rapid increase in the biomass amount. This trend was observed in all the fungal strains studied. Similarly, compared to media with 2% glucose, the presence of maltose or sucrose favored biomass formation and at the same time had a negative effect on enzyme activity. The ability of non-glucose sugars, such as sucrose, fructose, maltose, mannose, etc., to trigger catabolite repression has been reported for bacteria, yeasts, and fungi [39,40,41]. Like glucose, galactose represses β-galactosidase activity in *Escherichia coli* cultures [42].

The effect of glucose concentrations outlined two dose-dependent behaviors. The low concentrations (up to 0.5%) had a positive influence on sialidase activity by *P. griseofulvum* P29. In contrast, a higher glucose content (1, 2, 3, and 5%) resulted in the increased repression of enzyme activity. This means that catabolite repression is sensitive at glucose concentrations above 0.5%. These results were confirmed by the broth-spot assay (Figure 3). Similar differential responses to low and high glucose concentrations were reported for sialidase activity in *Clostridium perfringens* F4969, a human enteropathogenic strain [19]. The analysis revealed that high glucose concentrations primarily decrease the *nanI* gene expression, encoding exosialidase in most *C. perfringens* strains. A complete repression of all genes of the *nanAB* locus responsible for neuraminidase synthesis was described in cultures of *Streptococcus pneumoniae*, grown on glucose [17].

The importance of glucose concentrations for the catabolite repression has been also evaluated for *S. cerevisiae* [40,43,44] The authors suggested that transcription factor Mig1, which directly or indirectly controls the expression of hundreds of genes in yeasts, is responsible for the dose-dependent effect. While Mig1 is active at high glucose concentrations, the low glucose content causes inactivation of Mig1 by phosphorylation [45]. In filamentous fungi, the most important factor of CCR is the conserved transcription factor CreA/Cre1, a homologue of yeast *Mig1* [46]. Besides its primary function in CCR, CreA plays direct and indirect roles in diverse physiological processes, including secondary metabolism, iron homeostasis, N-glycan biosynthesis, oxidative stress response, unfolded protein response, and nutrient and ion transport [47]. At high glucose concentrations, CreA binds to a consensus sequence in the promoter, upstream of the start codon of the gene regulated by catabolite repression/activation. The first experimental piece of evidence about the nucleotide sequence of this consensus was performed for *P. canescens*. Chulkin et al. (2010) [48] found that CreA binds to the motif 5′SYGGRG 3′, where S is G or C, Y is C or T, and R is G or A (purine). Later, Penng et al. (2021) [49] showed that the binding site for CRE-1 in *N. crassa* and *A. niger* most probably is the consensus 5′TSYGGGG3′. Considering the sialidase gene of *P. griseofulvum*, the information about the putative promoter and CreA-binding site could be obtained after analysis of the full genome of the strain (NW_024467435.1) [50]. The sialidase gene has ID PGRI_091340, it is located between nucleotides 761,014 and 762 (one exon) and encodes an enzyme of 363 amino acids. Notably, the free software of Knudsen (1999) [51] for promoters search, Promoter-2.0, predicts the presence of a native promoter 500 bp upstream of the gene with high likelihood. Exactly 140 bp upstream of the ATG start codon of the sialidase gene the sequence 5′ CCTGGGG 3′ presents, being the consensus putatively involved in CreA binding.

The present study demonstrated also a decrease in glucose repression after 48 and 72 h of cultivation, particularly in the variants with 1 and 2% initial glucose content. In this period, almost complete uptake of glucose and de-repression of sialidase synthesis is established. However, repression was maintained in the variants with 3 and 5% initial concentrations because the residual glucose is at a relatively high level.

The present experiments with non-growing mycelia confirmed the glucose repression on sialidase activity. The *P. griseofulvum* P29 mycelia that are in a non-growing but metabolically active state [52,53] demonstrated the down-regulation of enzyme synthesis in the presence of high glucose concentrations. In addition, the results provide information that the functioning period of *P. griseofulvum* P29 mRNA for sialidase synthesis is not less than 6 h. Based on these results we used the non-growing mycelia to evaluate the effect of glucose-6-phosphate or other regulators of catabolite repression. The replacement of glucose by glucose-6-phosphate exerted also carbon catabolite repression in the tested strain. The reports on yeasts and fungi have shown that the glucose-6-phosphate is an effective promoter of CCR repressing genes responsible for the utilization of alternate carbon sources through gluconeogenesis and respiration [46,54]. The role of glucose-6-phosphate could be explained as an intermediator in gluconeogenesis for the synthesis of trehalose and glycogen [55]. De Assis et al. (2021) [56] assumed a regulatory complexity at both transcriptional and posttranslational levels that govern CreA in fungi.

Moreover, the present study demonstrated that the glucose repression on *P. griseofulvum* sialidase was regulated by the cAMP. Exogenously added cAMP (2, 5, and 8 mM) to fungal cultures incubated in the glucose-containing medium increased significantly the enzyme synthesis in comparison to the variants without cAMP. The effect of cAMP on sialidase synthesis has been very rarely studied. Hoyer et al. (1992) [57] reported a positive regulation of sialidase expression in *Salmonella typhimurium* LT2 by cAMP. However, it is known to have a de-repressive role in the expression of different enzymes in bacterial and fungal strains. High levels of cAMP can form a complex with cyclic AMP-receptor protein (CRP) that acts as an activator of a large number of catabolic genes [58]. cAMP-CRP complex positively regulates pectate lyase production in *Dickeya dadantii* see [59]. Several reports demonstrated the importance of cAMP-CRP for the transcription of numerous *Escherichia coli* genes [60]. Adnan et al. (2018) [46] noted cAMP as a crucial element of CCR in fungi. cAMP mediates the increased expression of hemicellulose and cellulase in *A. nidulans* [23]; cellulase in *Neurospora crassa* [61]; protease in *Metarhizium anisopliae* [62], etc. The addition of cAMP enhanced cellulolytic enzyme activity in *N. crassa* and *Trichoderma reesei* despite the presence of glucose in the medium [63].

The second finding revealed that the catabolite repression in the psychrotolerant strain *P. griseofulvum* P29 did not depend on temperature. Present results demonstrated that sialidase might be subject to catabolite repression by glucose both at 10 and 25 °C. Although the model strain was isolated from Antarctica, the step-down in temperature did not change the manner of glucose action. *P. griseofulvum* P29 sialidase was described as a cold-active enzyme [27]. Due to their unique characteristics, cold-active enzymes provoke increasing interest in scientific and industrial aspects. It is very important to understand the mechanisms regulating their synthesis, and in particular the effect of catabolite repression. However, very little is known regarding the effect of glucose on the synthesis of cold-active enzymes. Protease production by psychrotrophic microorganisms is affected by the presence or absence of some easily metabolizable sugars such as glucose and sucrose in the production medium [64]. Cold-active polygalacturonase expression in the fungus *Tetracladium* sp. drastically decreased in a medium supplemented with glucose compared to the pectin [65]. A similar effect has been reported for cold-active chitinase Chi21702 produced by the Antarctic bacterium, *Sanguibacter antarcticus* [66].

Third, we also observed that the expression of the sialidase gene participates in the regulation of enzyme synthesis by catabolite repression. The fungal cells grown on the glucose-containing medium showed a significant decrease in the gene expression level compared to the cells on the glucose-free medium. These data coincided with the decrease in sialidase activity as a result of glucose repression. After the depletion of glucose from the medium, recovery of gene expression was observed. It can be assumed that glucose represses sialidase synthesis at a transcriptional level.

## 5. Conclusions

The present study revealed the important role of the catabolite repression regulatory mechanism on sialidase synthesis by Antarctic fungal strain *P. griseofulvum* P29. To our knowledge, this is the first report on the effect of CCR on fungal sialidase and especially a cold-active sialidase. The repression by glucose was dose- and time-dependent. The replacement of glucose by glucose-6-phosphate promoted also CCR in *P. griseofulvum* P29, probably as an intermediator in gluconeogenesis. Exogenously added cAMP resulted in higher sialidase activity suggesting a de-repression via cAMP signaling in fungal cells. Although *P. griseofulvum* P29 is psychrotolerant and the sialidase produced by it is a cold-active enzyme, growth temperature does not affect the degree of catabolite repression. The drastic decrease in sialidase activity in the presence of glucose in the growth medium is consistent with a down-regulation of sialidase gene expression.

## Figures and Tables

**Figure 1 jof-10-00241-f001:**
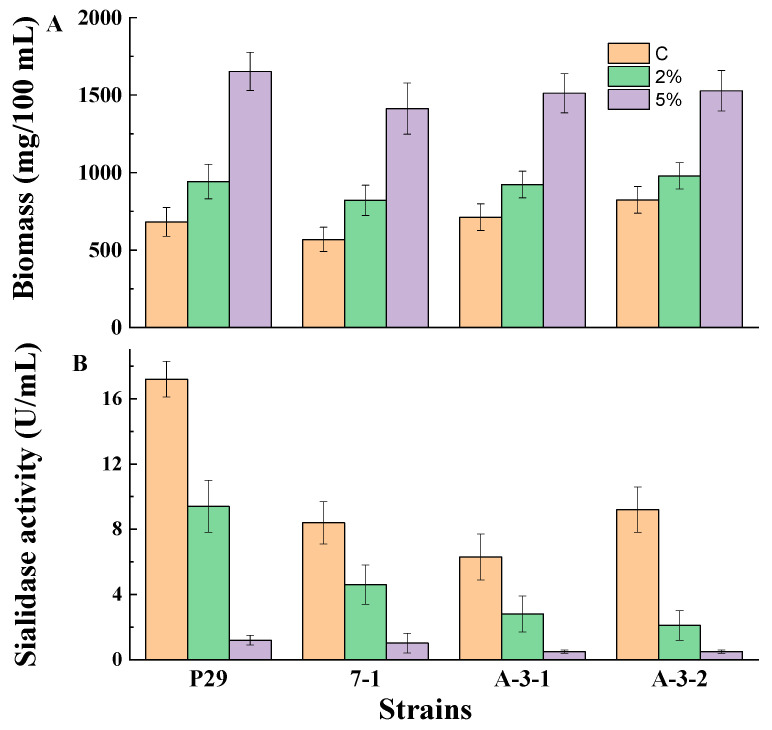
Biomass (**A**) and sialidase production (**B**) by fungal strains *P. griseofulvum* P29, *P. fimorum* III 7-1, *A. tubingensis* A 3-1, and *A. niger* A 3-2 grown on the medium supplemented or not with glucose. Values are means of three replicates; error bars represent the standard deviation. Glucose presence in the medium showed statistically significant effect on the growth and enzyme activity (*p* ≤ 0.05).

**Figure 2 jof-10-00241-f002:**
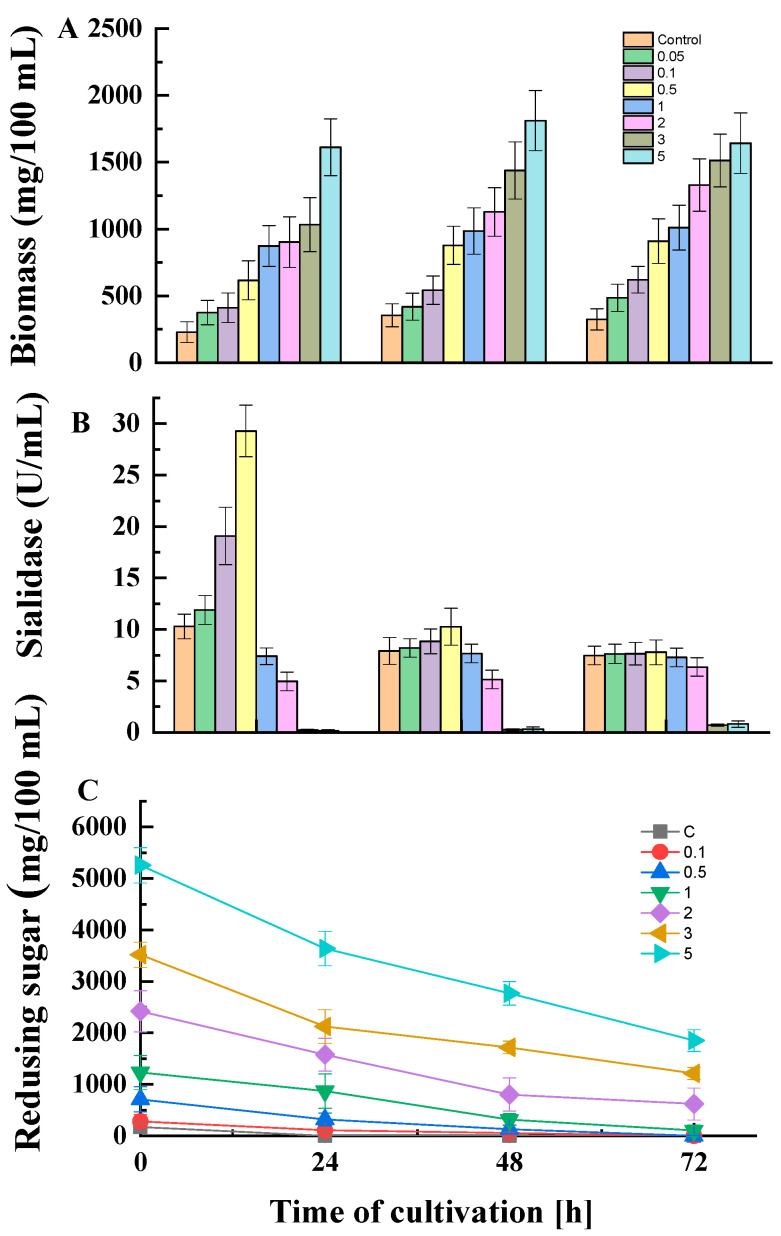
Dynamics of growth (**A**), enzyme activity (**B**), and residual substrate concentrations (**C**) in the absence or presence of different glucose concentrations values are means of three replicates; error bars represent the standard deviation. Glucose presence in the medium showed statistically significant effect on growth, substrate uptake, and enzyme activity (*p* ≤ 0.05).

**Figure 3 jof-10-00241-f003:**
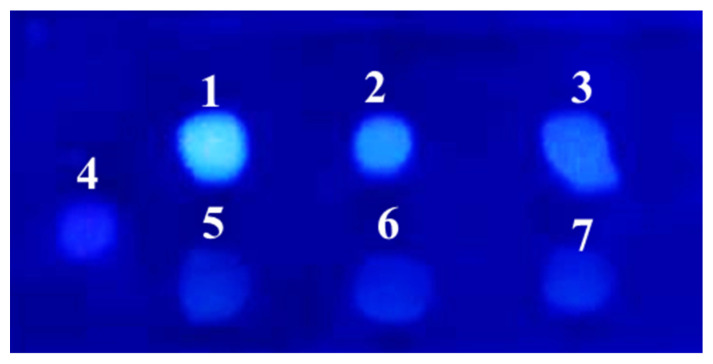
Sialidase activity analysis by the broth-spot assay of *P. griseofulvum* P29 strain grown in the absence or presence of glucose against the fluorescent substrate, 4MU-Neu5Ac: 1—Wh without glucose; 2—Wh + 2% glucose; 3—Wh + 5% glucose; 4, 5, 6, and 7—Negative controls (4—substrate 4MU-Neu5Ac alone; 5—Wh without glucose alone; 6—Wh + 2% glucose alone; 7—Wh + 5% glucose alone).

**Figure 4 jof-10-00241-f004:**
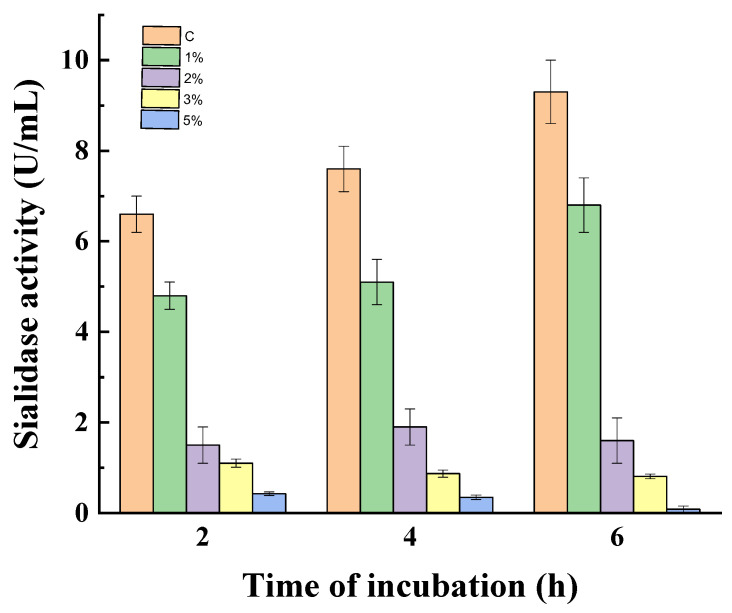
Glucose repression on sialidase synthesis in non-growing mycelia of *P. griseofulvum* P29. Values are means of three replicates; error bars represent the standard deviation. Glucose presence in the medium turns out statistically significant effect on the enzyme activity (*p* ≤ 0.05).

**Figure 5 jof-10-00241-f005:**
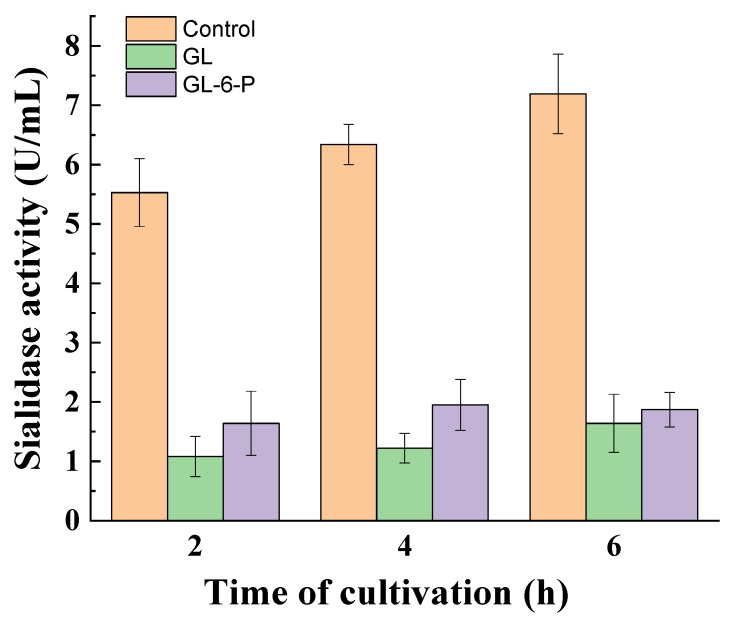
Sialidase activity in the medium supplemented with glucose or glucose-6-phosphate. Values are means of three replicates; error bars represent the standard deviation. The presence of glucose and glucose-6-phosphate in the medium showed a statistically significant effect on enzyme activity (*p* ≤ 0.05).

**Figure 6 jof-10-00241-f006:**
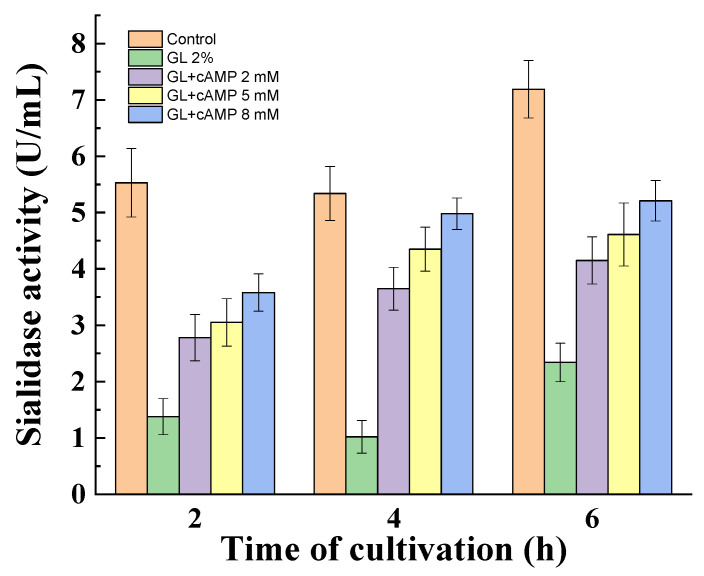
De-repression of sialidase synthesis in the addition of cAMP to the glucose. Values are means of three replicates; error bars represent the standard deviation. cAMP showed a statistically significant effect on enzyme activity (*p* ≤ 0.05).

**Figure 7 jof-10-00241-f007:**
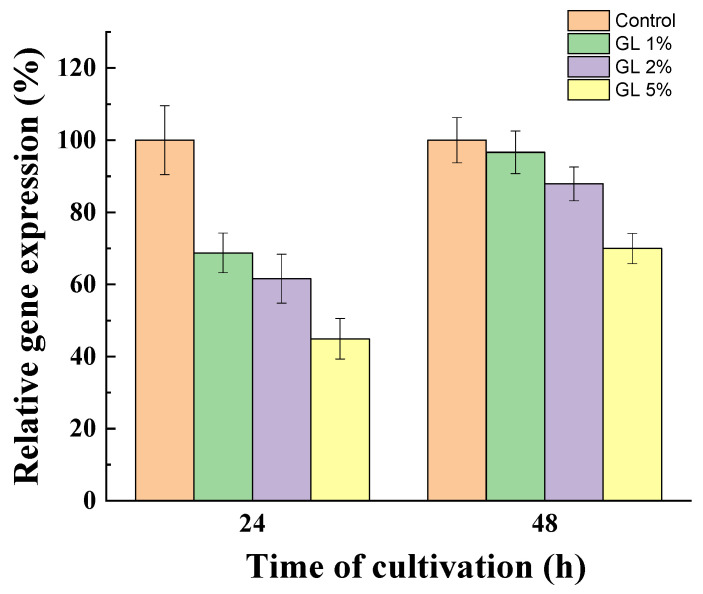
Relative expression of the sialidase gene in the presence of glucose compared to the control (without glucose in the medium), after 24 and 48 h of cultivation of *P. griseofulvum* P29. Values are means of three replicates; error bars represent the standard deviation. Glucose presence in the medium showed statistically significant effect on the sialidase gene expression (*p* ≤ 0.05).

**Figure 8 jof-10-00241-f008:**
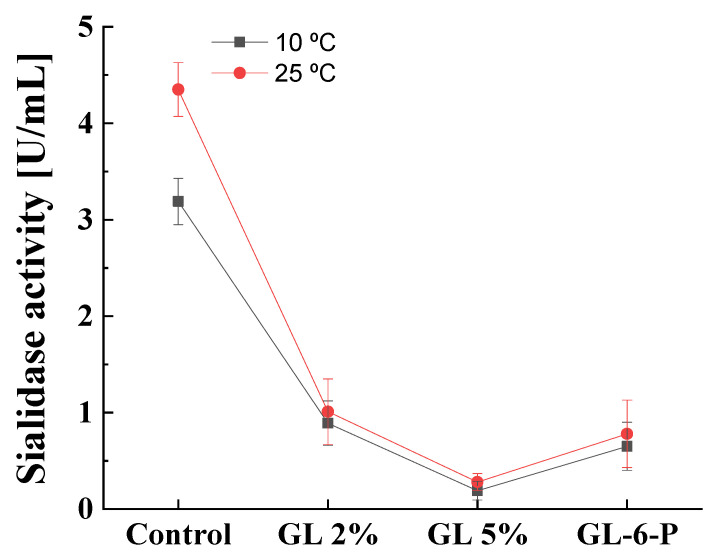
Glucose repression on cold-active sialidase at temperatures 10 and 25 °C. Values are means of three replicates; error bars represent the standard deviation. Growth temperature did not show statistically significant effect on the sialidase activity (*p* ≤ 0.05).

**Table 1 jof-10-00241-t001:** Primers used in RT-PCR experiments.

Primer	Sequence (5′-3′)	PCR Product (bp)	Reference
PenRTF	CAGAACTCTTCCGTTCGGCT	100	This study
PenRTR	TCACATAGGCTGCAAGGACG		
GADPH_F	CTGCTCTCTCATAGCCAACAC	157	[33]
GADPH_R	CTTCCTCCAATAGCAGAGGTTT		

**Table 2 jof-10-00241-t002:** Effect of the addition of different sugars (2%) to Wh on the growth and sialidase production by *P. griseofulvum* P29.

Carbon Source	Biomass [g/100 mL]			Sialidase Activity [U/mL]		
	24 h	48 h	72 h	24 h	48 h	72 h
Whey	0.2902	0.6438	0.7123	10.5	8.4	8.2
Wh + Glucose	0.8541	1.3481	1.7254	3.1	5.9	7.4
Wh + Maltose	0.6292	0.9245	0.9932	3.9	4.2	5.1
Wh + Sucrose	0.7998	1.1895	1.5932	2.5	6.7	4.3

## Data Availability

Data are contained within the article.

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
