# Peer review of "Glucose Catabolite Repression Participates in the Regulation of Sialidase Biosynthesis by Antarctic Strain Penicillium griseofulvum P29"

_jof, 2024, doi:10.3390/jof10040241_

Round 1
Reviewer 1 Report
The manuscrip t" Glucose Catabolite Repression Participates in the Regulation of 2 Sialidase Biosynthesis by Antarctic Strain Penicillium griseoful-3 vum P29" deals with a rather underinvestigated topic that is the regulation of sialidase production by fungal strains.
Althogh some conclusions of the work appear rather foregone, the work is well structred and the experimental set up and results clearly presented.
It seems that the research group have a very solid knowledge regarding the specific research field and therefore released an interesting manuscript.
Some minor flaws need to be amended (see the enclosed PDF file for details)
The manuscrip t" Glucose Catabolite Repression Participates in the Regulation of 2 Sialidase Biosynthesis by Antarctic Strain Penicillium griseoful-3 vum P29" deals with a rather underinvestigated topic that is the regulation of sialidase production by fungal strains.
Althogh some conclusions of the work appear rather foregone, the work is well structred and the experimental set up and results clearly presented.
It seems that the research group have a very solid knowledge regarding the specific research field and therefore released an interesting manuscript.
Some minor flaws need to be amended (see the enclosed PDF file for details)

Reviewer 2 Report
Dear authors, I hope the following observations are useful to you:
• Upload the sialidase gene of P. griseofulvum strain P29 to the databases and place the access code in the article. By not having this data, it is not possible to know that the gene they analyzed is the correct one.
• Is P. griseofulvum strain P29 sequenced or characterized previously?
• How many sialidase genes does this genus of fungi have? If there is more than one, are the oligos used specific to one of them or to all of them?
• The strains used mention P. griseofulvum Ð 29, P. fimorum ІІІ 7-1, Aspergillus tubingensis A 3-1, and A. niger A 3-2. Why was RT-PCR analysis not performed on all the strains used?
none
Round 2
Reviewer 2 Report
The observations made in the first revision have been answered and changes to the manuscript have been made. I have no more observations.
none